# Nematode Pheromones as Key Mediators of Behavior, Development, and Ecological Interactions

**DOI:** 10.3390/biom15070981

**Published:** 2025-07-09

**Authors:** Xi Zheng, Junjie Liu, Xin Wang

**Affiliations:** 1State Key Laboratory for Conservation and Utilization of Bio-Resources in Yunnan, Yunnan University, Kunming 650091, China; zhengxi1@mail.ynu.edu.cn (X.Z.); liujunjie1@stu.ynu.edu.cn (J.L.); 2School of Nursing, Yunnan University of Chinese Medicine, Kunming 650500, China

**Keywords:** nematode pheromones, ascaroside, glucoside, interspecies interactions, biosynthesis, biological control

## Abstract

Plant parasitic nematodes cause huge economic losses to agriculture and forestry every year, and chemical insecticides destroy the ecological environment. Researching the mechanism by which small-molecule signaling substances regulate nematode behavior and development is important for developing environmentally friendly biological control agents. Nematode pheromones are essential chemicals signaling intraspecies and interspecies communication, regulating development, reproduction, and social behavior. Their structural diversity enables ecological adaptation and cross-kingdom interactions, influencing fungal predation and plant immunity. This review focuses on the classification, function, and regulatory mechanisms of nematode pheromones, interspecific signal transmission, and biosynthesis pathways. We pay special attention to their potential as environmentally friendly biological control agents as well as the challenges currently encountered in their application.

## 1. Introduction

Chemical communication, a ubiquitous information transfer mechanism, is particularly critical for Nematoda [1]. Plant-parasitic nematodes impose substantial ecological and economic burdens, contributing to over USD 175.0 billion in annual crop losses worldwide through direct parasitism and soil ecosystem degradation [2]. As chemical signaling mediates diverse ecological interactions (e.g., pheromones [3,4,5], plant volatiles [6,7,8,9], microbial cues [10,11,12,13]), nematode-derived signaling molecules, especially ascarosides, have emerged as pivotal targets for sustainable pest control due to their functional versatility. These evolutionarily conserved molecules precisely modulate nematode behavior (e.g., aggregation, dispersal) and development (e.g., Dauer formation), as well as cross-species interactions (e.g., plant immune activation) [3,4,5,14], as shown in Figure 1. Deciphering the molecular mechanisms underlying these chemical signals will provide a crucial theoretical foundation for developing novel strategies in agricultural pest control, environmental restoration, and biomedical applications.

Nematodes orchestrate intra- and interspecies interactions through ascarosides, structurally diverse glycoside–lipid hybrids that function as multimodal signaling molecules. These compounds mediate critical ecological processes, population dynamics (e.g., ascr#3/5-driven aggregation vs. ascr#2-induced dispersal), reproductive strategies (male chemotaxis by ascr#3), and developmental plasticity (Dauer formation via DAF-7/TGF-β) [15,16]. Notably, plants and microbes exploit ascarosides for defense modulation (e.g., ascr#18-triggered jasmonate immunity), suggesting coevolutionary arms races [17,18,19]. The conservation of DAF/TGF-β and insulin pathways across species positions ascarosides as prime targets for sustainable nematode control and aging research.

In this review, we comprehensively examine nematode pheromones. We focus on their classification, functional roles, autoregulatory mechanisms, interspecies signaling, and biosynthesis pathways. Special attention is given to their potential as eco-friendly biocontrol agents, alongside the current challenges in their application. A key future research direction involves the identification of novel nematode signaling molecules and the investigation of their functions and regulatory mechanisms. Such efforts will expand the repertoire of known nematode pheromones and their regulatory networks. This will establish a discipline studying nematode signaling molecule-based biocontrol ecology and lay the foundation for developing environmentally friendly biocontrol formulations.

## 2. Nematode Pheromone-Mediated Communication in Ecosystems

Chemical communication releases signaling molecules (e.g., pheromones, secondary metabolites) that diffuse into the air, water, or soil, which are detected by other organisms and processed by them to elicit behavioral, physiological, or ecological responses [20]. This form of communication includes interactions between individuals and groups. Individual-level regulation involves mate selection, territorial marking, aggregation, and danger warnings, which involve signals such as sex pheromones, aggregation pheromones, and alarm pheromones [21]. Plant–insect or microbe–host interactions are common instances of interspecies communication, which plays an important role in the regulation of ecosystem functions [[22],,[23]]. Nematodes release glycoside pheromones to coordinate individual behavior and group interaction [24]. Chemical communication regulates predator–prey relationships, interspecies competition, and symbiotic associations. For instance, predatory fungi on nematodes intercept nematode-released pheromones to form infection structures that capture nematodes, thereby balancing fungal and nematode populations in soil habitats [25,26,27]. Similarly, when *Meloidogyne incognita* infect plants, the plants detect long-chain ascarosides released by the nematodes and metabolically edit them into clusters of short-chain ascarosides that inhibit nematode activity [19,28]. Chemical communication is of paramount importance for various organisms, from simple to complex.

## 3. Classification and Function of Nematode Pheromones

An important role of nematode pheromones is to regulate development, reproduction, social behavior, and population dynamics through small-molecule chemical signals produced and secreted by nematodes. They also serve as a language system for communication across species and between species [24,29,30,31], as shown in Table 1. Therefore, pheromone-mediated conserved chemical communication is critical for the survival and reproduction of nematodes [32].

### 3.1. Nematode Ascaroside Pheromones

Ascarosides are the primary constituents of nematode pheromones, representing a class of highly conserved small-molecule signaling compounds. Their chemical structure consists of a glycuronic sugar (3,6-dideoxy-L-ascarylose) linked to fatty acid-derived side chains of varying lengths, which are further modified with additional moieties derived from amino acids, folate, and other primary metabolites, forming a diverse and extensive signaling system (Figure 2). To date, more than 200 distinct ascarosides have been identified across over 20 species of free-living and parasitic nematodes, with their biological functions spanning developmental regulation and social behaviors [17,47]. The chemical composition, structural diversity, and relative abundance of ascarosides vary significantly depending on developmental stage, lifestyle, and species [48,49,50]. The model organism *Caenorhabditis elegans* has been instrumental in elucidating the nematode pheromone network. Comparative genomics and metabolomics studies have revealed that the modular assembly of ascaroside derivatives endows them with functional diversity in biological activity [51]. In addition, mammals, plants, and microorganisms share a biosynthetic pathway that produces ascaroside-like signals. These organisms can also diversify the nematode ascaroside signaling system, generating structurally distinct variants [52].

### 3.2. Nematode Glucoside Pheromones

As well as ascaroside pheromones, nematodes can produce hundreds of glucosides by conjugating different sugar molecules (e.g., glucose, galactose) with certain fatty acids. Representative examples include ABZ–glucoside generated during xenobiotic detoxification [41], indole–glucoside produced from bacterial indole metabolism [42], and phenazine glucoside synthesized in response to the pathogenic toxin 1-hydroxyphenazine (1-HP) from *Pseudomonas aeruginosa* [43]. The glycosylation-mediated conjugation of xenotoxic small molecules into glycoside derivatives constitutes a crucial survival strategy for nematodes. Recent metabolic and biosynthesis studies demonstrate that *C. elegans* and other nematodes employ a streamlined assembly strategy involving ester/amide bond formation. This is performed to generate numerous modular glucoside metabolites, using glycosides as central scaffolds. Some of these compounds have been functionally validated as signaling molecules that modulate nematode behavior and development [53].

### 3.3. Other Nematode Pheromones

In recent years, several non-glycosidic compounds have been identified as pheromones, including dafachronic acids, unsaturated fatty acids, and other small-molecule metabolites [54,55]. Dafachronic acids are derived from 3-keto-cholenoic acid and feature a terminal carboxylic acid group on the sterol side chain, structurally resembling bile acids and steroid hormones. These compounds function as ligands for the DAF-12 nuclear receptor, playing a pivotal role in regulating nematode life cycles, particularly in Dauer larva formation and recovery. Thus, they represent a class of hormone-like signaling molecules critically involved in developmental and behavioral modulation [44].

Emerging evidence indicates that unsaturated fatty acids (UFAs), primarily derived from the lipid metabolism and peroxisomal β-oxidation pathways, actively participate in nematode lifespan modulation [45]. Nematodes also secrete amino acid-conjugated long-chain fatty acids as signaling molecules in response to dynamic social environments. Male nematodes (and to a lesser extent, hermaphrodites) secrete nacq#1—a small-molecule signaling substance belonging to the N-acylated glutamine class—which regulates a conserved gene regulatory network via chemosensory pathways and nuclear hormone receptors (NHRs). This signaling cascade counteracts diapause, accelerates developmental progress, and promotes sexual maturation, albeit at the cost of a reduced lifespan. At concentrations ranging from picomolar (pM) to low nanomolar (nM), nacq#1 significantly enhances reproductive development, particularly during the transition from the L4 larval stage to young adulthood (YA), resulting in a ~30% increase in egg-laying output on the first day of reproduction [46]. Notably, nacq#1 exhibits an antagonistic relationship with ascaroside pheromones (e.g., acr#2 and acr#3) in governing nematode reproductive strategies: while nacq#1 promotes accelerated reproductive, Ascarosides induce developmental arrest (Dauer formation). Together, these small-molecule systems fine-tune the Dauer development trade-off, ensuring adaptive responses to environmental cues. Intriguingly, nacq#1 is evolutionarily conserved and detected across diverse animal phyla, including human sweat. This suggests that amino acid-derived metabolites may serve as universal chemo-effectors to modulate physiological processes. This functional conservation implies that such signals likely operate through specialized neural circuits. This highlights potential parallels in the inter- and intra-tissue communication mechanisms between vertebrates and invertebrates. Furthermore, nematodes employ fatty acid metabolism and the mevalonate pathway to biosynthesize an array of non-glycosylated small-molecule signals, including fatty acids, terpenoids, and peptides. These compounds are typically labile and volatile, enabling rapid response coordination in dynamic environments [56].

Nematodes exhibit remarkable genome size variation (tens of Mb to >1 Gb), with plant-parasitic species typically exceeding 80 Mb while *C. elegans* maintains a compact ~100 Mb genome [57,58]. Hundreds of signaling molecules—particularly ascarosides—have been identified via MS/MS and GC-EI/MS. Their bioactivity is finely modulated by structural variations (e.g., side chain length, ascarylose linkage, and terminal modifications, as seen in ascr#18-ascr#25 and hbas#3/mbas#3) and they typically function as concentration-dependent mixtures [59,60,61]. Despite the genomic complexity observed in nematodes and other model organisms (*Drosophila*, mice), known signaling chemicals remain limited, underscoring the need for integrated comparative genomics, metabolomics, and trace compound isolation to systematically unravel nematode pheromone systems.

## 4. Regulatory Functions and Mechanisms of Nematode Pheromones

Nematode pheromones, characterized by highly conserved chemical structures, serve as essential signaling molecules for inter- and intraspecies communication [30,40]. These pheromones exert multifaceted regulatory effects throughout the nematode life cycle, modulating behaviors such as aggregation, avoidance, developmental progression, and reproduction [24,61].

### 4.1. Regulation of Nematode Behavior by Pheromones

The chemical modulation of nematode social behaviors—including chemotaxis, avoidance, aggregation, dispersal, and foraging—represents a critical research focus in neurobiology and chemical ecology, with significant implications for pest control strategies [62,63]. Chemotactic interactions, such as attraction and repulsion, are mediated by 12 pairs of specialized chemosensory neurons in *C. elegans*’ head, as shown in Table 2 [64,65]. Each pair expresses distinct receptors that recognize specific pheromone molecules, activating downstream signaling cascades to generate appropriate behavioral outputs, as shown in Figure 3 [65,66]. Among nematode pheromones, ascr#8 has been identified as the principal male-attracting chemosignal [37]. Notably, nematodes employ complex behavioral strategies involving synergistic effects among multiple pheromone components, demonstrating the functional versatility of these chemical mediators. The ascaroside pheromones (ascr#2, ascr#3, and ascr#4) exhibit potent male attractant activity at sub-Dauer-inducing concentrations, with this response mediated by the cephalic chemosensory neurons ASK and CEM [55]. Genetic evidence indicates that mutations in NPR-1, encoding a neuropeptide Y receptor homolog, enhance nematode responsiveness to specific Ascarosides while attenuating aversive behaviors [67]. The RMG interneuron functions as a central processing unit in NPR-1-mediated signaling by integrating multiple sensory inputs to differentiate between attractive and aggregative chemical cues, depending on molecular specificity and concentration profiles [68].

Moreover, both food availability and TGF-signaling perturbations can drive aggregation behavior in nematodes. The Indoleascarosides (icas#3, icas#9, icas#10, icas#1, icas#7) function as pheromonal regulators of this social behavior [35,37]. Notably, icas#3 exhibits strong attraction toward hermaphroditic nematodes at exceptionally low concentrations (1 pM). Structurally, icas#3 closely resembles ascr#3 but contains an additional indole moiety derived from tryptophan. Despite this similarity, their behavioral effects differ significantly: while ascr#3 attracts male nematodes, it repels hermaphrodites, highlighting distinct sex-specific regulatory roles [16]. Both indole glycosides and indole ascarosides modulate nematode behavioral responses through the NPR-1 receptor, further underscoring its pivotal role in pheromone-mediated neuroregulation [68].

Amphid single-ciliated chemosensory neuron type K (ASK) plays a critical role in mediating behavioral responses in *C. elegans*. Previous studies have demonstrated that ASK neurons are essential for behavioral responses to non-indole ascarosides in both males and hermaphrodites [68]. ASK sensory neurons form gap junction connections with the RMG interneuron. This interneuron functions as a central hub to integrate inputs from ASK and other sensory neurons to regulate aggregation and related social behaviors [69]. However, icas#3-mediated behavioral modulation operates independently of RMG but requires the AIA interneuron. Thus, the ASK-RMG circuit transduces ascr-type ascarosides, while the ASK-AIA pathway processes indole-modified ascarosides to govern attraction and aggregation behavior.

### 4.2. Dauer Larva Induction and Regulatory Mechanisms

Each nematode undergoes four distinct larval stages, with each transition marked by the development of a new cuticle and the molting of the previous cuticle [70,71]. Dauer larvae constitute an evolutionarily conserved developmental adaptation that has emerged as a specialized survival and dispersal strategy within the Rhabditida order and numerous parasitic nematode species [72,73]. This metabolically quiescent, development-arresting state is triggered by environmental stressors, including nutrient deprivation, thermal stress, or high population density [74]. The Dauer phase of nematodes is characterized by profound physiological changes, including changes in metabolism, cell stress tolerance, and lifespan extension. Collectively, these adaptations allow long-term persistence (spanning months or longer) under hostile conditions until favorable environmental cues reactivate development [75].

The regulation of Dauer entry and exit remains a central focus in developmental biology, and inhibiting these processes serves as a crucial target for nematode control [73]. Besides unfavorable environmental cues, ascarosides also play a vital role in Dauer formation. The concept of nematode Dauer pheromones was proposed by Golden and Riddle in 1982 [76], but its chemical identity was not determined until 2005 when the bioactivity-guided purification of ascr#1 revealed its identity. However, its active concentrations were toxic [33]. By 2008, ascr#2 and ascr#3 were demonstrated to exhibit potent Dauer-inducing activity at nM–µM levels, with a 10 nM equimolar mixture significantly increasing Dauer formation in populations [33,77]. Furthermore, ascr#5 synergizes with ascr#2 and ascr#3 to enhance Dauer induction [34,36,76]. Notably, while ascr#2 and ascr#3 have also been identified in other nematodes (e.g., *Caenorhabditis briggsae*), only ascr#2 is required to trigger Dauer formation in this species [78].

Nematodes’ entry into the Dauer stage involves complex interactions between multiple signaling pathways. These interactions are primarily triggered by insulin and TGF-signaling downregulation to initiate Dauer development [18]. Under environmental stressors such as crowding, food limitations, or elevated temperature, the chemosensory neurons of ASK detect changes in the pheromones ascr#2 and ascr#3 via a pair of pheromone receptors, SRBC-64 (sterpentine receptor, class BC-64) and SRBC-66, located at the ciliary tips. These receptors activate the inhibitory G protein GPA-3, which interacts with membrane-associated guanyl cyclase DAF-11 to convert GTP into cGMP. The TAX-2/TAX-4 cyclic nucleotide-gated (CNG) ion channels then transduce cGMP levels into an ionic influx, leading to membrane depolarization and subsequent Ca^2+^ and Na^+^ entry. This cascade activates downstream endocrine signaling pathways that modulate TGF-β and insulin-like signaling [79]. In *C. elegans*, mutations in srbC-64, srbC-66, or gpa-3 disrupt the nematode’s ability to respond to ascaroside pheromones (C6, C9, C7) and enter the Dauer stage. Since daf-7 (a TGF-β ortholog) negatively regulates Dauer formation, exposure to Dauer-inducing pheromones downregulates the expression of the GPCR str-3 in ASI neurons [80,81]. Thus, the G protein GPA-3 signaling pathway orchestrates the transcriptional regulation of downstream genes governing Dauer formation. Downstream from pheromone signaling, reduced cGMP levels suppress the production of insulin-like peptides (ILPs) and TGF-β, promoting the nuclear translocation of DAF-16/FOXO and the DAF-3/DAF-5/SMAD complex. This inhibits the biosynthesis of steroid ligands for DAF-12, a nuclear hormone receptor homologous to mammalian steroid receptors. Consequently, the unliganded DAF-12 binds to its corepressor DIN-1/SHARP, activating Dauer-specific gene expression while directly or indirectly suppressing hormone biosynthesis-related genes. This transcriptional reprogramming ultimately prevents development at the Dauer stage [18]. Furthermore, Dauer’s entry and exit are regulated by steroid hormone signaling pathways, which are crucial for fully exiting developmental arrest and actively promote this process. Notably, the spatiotemporal dynamics of steroid hormone regulation during exit from developmental arrest resemble those observed during entry into developmental arrest [82].

### 4.3. Pheromonal Regulation of Nematode Mating Behavior

Ascarosides serve as key chemosensory signals mediating intra- and interspecific communication in nematodes, which are primarily responsible for sexual recognition. An ascr#2, ascr#3, and ascr#4 mixture at ultra-low femtomolar (fM) to picomolar (pM) concentrations attracts males in the nematode population [16,83]. The relative abundance of these ascaroside molecules varies significantly, with individual compound assays revealing that ascr#3 exhibits stronger male-attracting activity than ascr#2. Interestingly, this contrasts sharply with their roles in Dauer formation. Ascr#2 demonstrates higher potency than ascr#3 in inducing Dauer developmental arrest. Conversely, male nematodes attract hermaphrodites through indole carboxyl (IC)-modified ascarosides (ICAS#). Structurally, these IC-ascarosides are characterized by an indole-3-carbonyl moiety conjugated to the C-4 position of the ascarylose sugar. Behavioral assays demonstrated that icas#1, icas#3, and icas#9 effectively elicit hermaphrodite attraction, with icas#3 and icas#9 exhibiting particularly pronounced effects at 100 fM concentrations [16]. Furthermore, IC-ascarosides (fM-pM) significantly enhance aggregation behaviors in both hermaphroditic and male nematodes.

The male-attracting effects of ascr#2, ascr#3, and ascr#4 are mediated by two specific sensory neurons: the cephalic companion neurons (CEM) and the amphid single-cilium sensory neuron K (ASK) in males. As previously described, ASK neurons regulate Dauer entry through G protein-coupled signaling, while indoleascarosides modulate nematode aggregation behaviors via responses in ASK and its downstream interneuron AIA. Thus, nematodes coordinate alternative behavioral outputs—reproduction vs. developmental arrestthrough shared chemosensory pathways that process overlapping ascaroside signals [16]. This neural mechanism enables the context-dependent modulation of population dynamics by the same family of signaling molecules.

In response to *P. aeruginosa*, the hermaphroditic *C. elegans* AWA sensory neurons are inhibited, leading to str-44 induction within these neurons. The AWA neurons are regulated by the transcription factor zip-5, which is upregulated by the pathogen. STR-44 is a pheromone receptor that inhibits the avoidance of pheromones ascr#2, ascr#3, and ascr#5. Its activity in AWA neurons reduces pheromone responses, increasing the likelihood of mating with males [84].

### 4.4. Pheromone-Mediated Modulation of Nematode Lifespan

*C. elegans* lifespan extension is regulated through the DAF-16/insulin signaling pathway and the DAF-12 pathway. However, the Schroeder research team introduced the concept of ascr-mediated increase in lifespan (AMILS), demonstrating that the ascaroside receptor DAF-37, upon activation by the Dauer phenomone ascr#2, extends nematode lifespan by nearly 20% through a Srtuin-dependent mechanism [85]. Unlike the insulin/IGF-1 pathway, which governs Dauer formation, AMILS promotes longevity without compromising reproductive fitness or feeding rates. This suggests a distinct regulatory mechanism. Additionally, the N-acyl glutamine-derived metabolite nacq#1, excreted by nematodes, accelerates reproductive development in *C. elegans* by hastening sexual maturation in hermaphrodites and shortening lifespan. Intriguingly, this nacq#1-induced developmental acceleration depends on chemosensory perception mediated by conserved nuclear hormone receptors (NHRs). Notably, structurally similar fatty-acylated amino acid derivatives have been identified in both vertebrates and invertebrates, implying that related signaling compounds may function as evolutionarily conserved endocrine regulators across metazoans [46].

## 5. Nematic Pheromones Mediate Interspecies Interactions

Beyond functioning as “self-beneficial compounds” to regulate nematode development and stress resistance, accumulating evidence reveals that nematodes employ highly conserved ascaroside pheromones for “cross-kingdom communication”, predominantly occurring in soil ecosystems where they interact with other organisms, such as nematode–fungi [86], nematode–bacteria [87,88], and nematode–plant systems [89,90,91]. Consequently, through long-term coevolutionary interactions, plants, animals, and microorganisms have developed conserved genetic mechanisms to perceive and respond to nematode pheromones, playing a pivotal role in maintaining soil communities’ ecological equilibrium [92].

### 5.1. Nematode Pheromones Modulate the Interaction Between Nematodes and Fungi

In ecosystems, predator recognition of specific pathogens or food resources plays a crucial role in predator–prey coevolution. Nematophagous fungi and nematodes represent a well-documented interaction system in soil environments. With over 380 reported species, nematophagous fungi are widely distributed and have evolved a conserved coexistence with ubiquitous nematodes [93]. These fungi employ diverse predatory structures—such as adhesive networks, constricting rings, and sticky knobs—to capture and digest nematodes for nutrients [94,95]. Nematode-derived ascarosides, a class of small-molecule pheromones, have been identified as key signaling molecules that induce fungal predatory structures. Through long-term coexistence with nematodes, fungi have evolved specialized molecular recognition mechanisms to detect nematode presence and initiate trap formation. *Arthrobotrys oligospora*, a model nematophagous fungus, responds to live nematodes and specific ascarosides (e.g., ascr#5, ascr#7, ascr#18) by producing adhesive networks for nematode capture [27]. Notably, both native and parasitic nematodes secrete ascarosides [40], suggesting that fungal–nematode interactions represent an evolutionarily conserved paradigm, offering foundational insights into nematode interactions with other organisms [96].

In two predacious fungi species (*A. oligospora* and *Arthrobotrys flagrans*), the mechanism by which G protein-coupled receptors (GPCRs) respond to nematode pheromones to trigger trap formation has been elucidated. In *A. flagrans*, the GPCR GprC exhibits dual localization on both the plasma membrane and the outer mitochondrial membrane. Mitochondrial GprC enhances respiratory metabolism. Upon nematode presence, GprC binds to the G protein α subunit GasA, initiating downstream signaling to activate fungal predatory behavior. Notably, spatial heterogeneity exists in GprC-GasA interactions: at the hyphal tip, their interaction primarily occurs on mitochondrial membranes, whereas at the hyphal base, it predominantly occurs on the plasma membrane. This spatial regulation may reflect functional specialization across the hyphal regions. Under nematode exposure, the GprC-GasA pathway suppresses the expression of the nuclear gene artA, inhibiting the production of the secondary metabolite 6-methylsalicylic acid (6-MSA). Concurrently, nematodes induce an increase in intracellular ROS levels and oxygen consumption, a process dependent on the GprC-GasA axis. Enhanced respiratory metabolism via this pathway likely provides energy for fungal predation. Unlike typical G protein signaling that relies on downstream MAPK or cAMP-PKA central pathways, the GprC-GasA pathway operates independently. Intriguingly, *A. flagrans* GprC shares conserved ascr# (ascarosides)-binding motifs with nematode GPCRs (SRBC64 and SRBC66). Functional complementation experiments demonstrated that replacing the nematode receptor’s last three transmembrane domains with those of GprC restored predation defects in GprC-deficient fungi, suggesting convergent evolution between fungal and nematode GPCRs [97,98].

The nematophagous fungus *A. oligospora* detects nematode-specific pheromones (ascarosides) through specialized G protein-coupled receptors (Gpr1-like GPCRs), triggering the formation of predatory structures. Concurrently, the additional 69 Pth11-like GPCRs in this fungus sense other small-molecule signals derived from nematodes. Upon exposure to *C. elegans*, several GPCRs are upregulated, particularly prior to trap development. Specifically, the glucose-sensing receptors Gpr2 and Gpr3 form a complex with the Gα subunit Gpa2, activating the cAMP-PKA signaling pathway to regulate trap morphogenesis. This intricate fungal–nematode interaction system reveals a sophisticated cross-kingdom communication mechanism, wherein fungal GPCRs serve as key mediators of predator–prey recognition. These findings highlight the potential for targeting nematophagous fungal GPCRs in developing novel biocontrol strategies against parasitic nematodes [25]. Moreover, *A. oligospora* produces various volatile compounds that attract nematodes, such as methyl 3-methyl-2-butenoate (MMB), which interferes with the mating of nematodes. These volatile compounds are primarily recognized by G protein-coupled receptors (GPCRs) expressed in nematode AWCs [99].

### 5.2. Regulation of Nematode–Pathogen Interactions by Nematode Pheromones

Nematode pheromones play a crucial regulatory role in mediating interactions between nematodes and pathogenic microbes. For instance, *P. aeruginosa* PA14, upon infecting nematodes, induces the expression of the chemoreceptor STR-44 in AWA sensory neurons, thereby suppressing the nematode’s avoidance response to ascarosides (e.g., ascr#2, ascr#3). STR-44 functions as a pheromone receptor, with its expression regulated by the transcription factor ZIP-5 and the TRPV channel OCR-2, ultimately modulating behavioral responses. Interestingly, exposure to PA14 shifts nematode behavior from pheromone avoidance to attraction, promoting mating behavior—an effect potentially linked to pathogen virulence, as attenuated strains (e.g., PA14-ΔgacA) exert weak modulation [84]. Beyond pheromone detection, nematodes also sense pathogen-derived volatile compounds (e.g., 2-heptanone) via AW-Con neurons, activating the receptor STR-2. This process relies on PLC-mediated signaling rather than the canonical TRPV or cGMP-dependent pathways, demonstrating an alternative mechanism for pathogen avoidance behavior [100].

Pathogen-derived signaling molecules mediate transgenerational immune priming in nematodes. Parental worms perceive volatile compounds such as cyanide released by pathogenic bacteria, which activates the enzyme CYSL-2 to convert cyanide into β-cyanoalanine. This metabolite functions as a transgenerational signal transmitted to offspring, where it activates the transcription factors MDT-15 and SKN-1, thereby enhancing the expression of stress-responsive genes (e.g., gst-4, cysl-2) and improving offspring’s resistance to pathogens [101].

### 5.3. Nematode Pheromones Modulate Nematode-Plant Interactions

In plant–parasitic nematode interactions, ascarosides, present in phytoparasitic nematodes, can trigger immune responses in crops such as maize, rice, wheat, and soybean. Plants employ pattern recognition receptors (PRRs) to detect nematode-associated molecular patterns (NAMPs) invading nematodes. When the plant roots or leaf tissues are exposed to ascr#18 during nematode infection, this molecule acts as a defense signal that primes plant resistance against pathogen threats. Notably, ascr#18, which is highly abundant in many phytoparasitic nematodes, is currently the only reported plant-associated molecular pattern (pAMP) in this context. The leucine-rich repeat receptor-like kinase NILR1 (NEMATODE-INDUCED LRR-RLK 1) serves as the immune receptor for ascr#18 [102], mediating defense responses in both monocots and dicots (e.g., *Arabidopsis*, tomato, potato, and barley). Ascr#18-induced signaling enhances resistance against pathogens and pests, including root–knot interactions and cyst nematodes, by activating hallmark defense mechanisms. Upon perception, ascr#18 elicits canonical plant immune pathways, such as MAPK cascades, SA/JA phytohormone signaling, and defense-related gene expression [19,103]. This response strengthens plant resistance against diverse pathogens, including viruses, fungi, bacteria, oomycetes, and nematodes [104].

Notably, both microbial communities and host plants possess evolutionary-conserved capacities to perceive and catabolize nematode pheromones. Genetic evidence demonstrates that *Arabidopsis* metabolizes ascr#18 through the peroxisomal β-oxidation pathway, converting it into structural derivatives (e.g., ascr#9) that exhibit potent nematode-repellent activity [19]. Intriguingly, while unidentified proteinaceous components in nematode culture filtrates elicit similar defense responses, their chemical identities remain uncharacterized. Functional studies confirm that these unknown elicitors predominantly activate immunity through the leucine-rich repeat receptor-like kinase (LRR-RLK) pathway, with NILR1 serving as the central receptor module [103]. These findings collectively establish a mechanistic framework for developing pheromone-based biocontrol strategies against phytoparasitic nematodes [19,52,103,105].

### 5.4. Other Aspects of Communication Through Nematode Pheromones

Most parasitic nematodes possess an infective juvenile stage (IJ or IL3), which is critical for survival outside of the host. In *Bursaphelenchus xylophilus*, the L3 larvae produce ascarosides (asc-C5 and asc-ΔC6) that promote beetle pupation upon reaching the adult stage; the beetles synthesize fatty acid ethyl esters, which facilitate the nematodes’ development into fourth-stage (LIV) larvae. Notably, adult beetles secrete specific ascarosides (particularly asc-C9, ascr#10) that attract LIV nematodes, potentially inducing their entry into the beetle tracheal systems for subsequent transmission to new pine hosts [106].

Originally recognized as pheromone-like signaling molecules in nematodes, ascarosides have now been identified in diverse higher organisms, including the Japanese pine sawyer (*Monochamus alternatus*) and mice, highlighting their broad functional significance. Studies on *B. xylophilus* control revealed that the vector beetle *M. alternatus* produces structurally distinct ascarosides (e.g., asc-C9), synthesized via β-oxidation, which synchronize the developmental cycles of the nematode and beetle, reinforcing their symbiotic relationship. This coordination contributes to the widespread dispersal and epidemic outbreaks of pine wilt disease in China []. Furthermore, ascarosides delay beetle development during winter, aiding overwintering survival. In addition, they regulate the life history alignment of *M. alternatus* and *B. xylophilus*, exacerbating disease transmission [107,108].

Although ascarosides are highly conserved as small-molecule signals in nematodes, recent studies have uncovered their biosynthesis in plants, animals, and microbes, expanding their ecological and biological relevance. In plant-parasitic nematode interactions, ascarosides trigger immune responses in crops such as maize, rice, wheat, and soybean. Exposure of root or leaf tissues to ascr#18, a molecule analogous to microbial PAMPs, rapidly activates pattern-triggered immunity (PTI) via membrane-bound receptors like *NEMATODE-INDUCED LRR-RLK1 (NILR1)*, triggering MAPK cascades, SA/JA signaling, and defense-related genes, a defense strategy exploited in *Arabidopsis* and tomato [19,103]. Plants metabolically process long-chain ascr#18 via β-oxidation to generate short-chain ascaroside clusters (ascr#10, ascr#1, and ascr#9), which enhance systemic immunity against pathogens such as Turnip crinkle virus, *Pseudomonas syringae* and PV, as well as tomatoes, fungi (*Blumeria graminis* f. sp. hordei, *Phytophthora infestans*), and nematodes (*Heterodera schachtii*, *Meloidogyne incognita*) []. These findings offer novel strategies for sustainable agriculture by reducing reliance on harmful agrochemicals, with implications for global food security.

Ascarosides govern diverse nematode behaviors (e.g., aggregation, avoidance) and developmental processes. Recent research employing optogenetics, RNAi screening, and genetic validation demonstrates that nematode pheromones can remodel neurodevelopment. The two most abundant pheromones, ascr#3 and ascr#10, are detected by chemosensory neurons ASK and ASI via specific receptors, respectively. These neurons subsequently release neuropeptides and glutamate, integrating signals into the interneuron AIA. This cascade suppresses nuclear translocation of the DAF-16/FOXO transcription factor in AIA, which then secretes insulin-like peptide INS-1 to non-autonomously activate the insulin signaling pathway across neurons, inhibiting macroautophagy and accelerating neurodegeneration. Notably, brief exposure to pheromones during juvenility primes adult nematodes for insulin pathway activation, marked by diminished autolysosome formation and impaired macroautophagy [109]. These insights provide a framework for understanding human neurodegenerative disorders, such as Alzheimer’s and Parkinson’s diseases.

## 6. Nematode Pheromone Biosynthesis

The biosynthesis of the two major types of nematode pheromones—ascarosides and glucosides—follows a similar modular assembly pattern: ascarylose, paratose, and glucose serve as core scaffolds, to which small molecules derived from primary metabolism (e.g., lipids, amino acids, neurotransmitters, and nucleotides) are attached [53,108]. Diverse combinatorial arrangements of these modules generate hundreds of structurally distinct ascarosides and glucosides [51]. A key advantage of modular assembly lies in its efficiency: a limited number of transferase genes or conserved biosynthetic pathways can produce a vast library of small-molecule compounds, enabling these metabolites to serve as versatile signaling molecules [110]. Identifying novel variants of primary metabolic derivatives that function as critical signals in *C. elegans* and other nematodes will provide a powerful impetus for comprehensively analyzing metabolism in higher animals.

### 6.1. Biosynthesis of Nematode Ascaroside

Ascaroside pheromone biosynthesis demonstrates remarkable evolutionary conservation, relying on the peroxisomal β-oxidation (PβO) pathway involving four key enzymes—ACOX-1, MAOC-1, DHS-28 and DAF-22 [17,31,40,61]—as shown in Figure 4. Among the primary fatty acid degradation pathways in eukaryotes, PβO generates metabolic byproducts, including heat, NADH, and either acetyl-CoA or propionyl-CoA. In nematodes, the synthesis begins with long-chain fatty acid precursors (approximately C30) that are progressively shortened through iterative cycles of PβO to generate ascaroside derivatives with C3-C11 side chains [111]. The biochemical pathway initiates with ACOX-1 (acyl-CoA oxidase) catalyzing the formation of α, β-unsaturated bonds in long fatty acid chains. This is followed by MAOC-1 (2-enoyl-CoA hydratase) mediating the hydration of these unsaturated bonds, DHS-28 (β-hydroxyacyl-CoA dehydrogenase) producing β-hydroxyacyl-CoA intermediates, and finally DAF-22 (thiolase) completing each β-oxidation cycle [39,112,113,114]. Each full cycle results in the removal of two carbon units from the fatty acid side chain through coordinated enzymatic actions: ACOX enzymes (functioning as homo- or heterodimers) convert α-β single bonds into double bonds; MAOC-1 hydroxylates the resultant double bonds; DHS-28 oxidizes β-hydroxyacyl-CoA to β-ketoacyl-CoA; and DAF-22 cleaves the thioester bond to terminate the cycle. The *C. elegans* genome contains seven ACOX genes (cel-acox-1.1 to cel-acox1.6 and cel-acox-3) that display distinct substrate specificities. The ACOX-1.1 homodimer preferentially acts on ascarosides with side chains of C9 or longer, while the ACOX-1.1/ACOX-3 heterodimer processes shorter chain substrates (C7 or less). The ACOX-1.2 homodimer regulates ω-ascaroside production for side chains containing fewer than 5 carbons [[114],]. An interesting comparative observation comes from the Japanese pine sawyer beetle (*M. alternatus*), where the multifunctional enzyme Mfe2 simultaneously performs the biochemical roles of both MAOC-1 and DHS-28 [108]. Despite significant progress in understanding ascaroside biosynthesis, critical questions remain unanswered regarding the biosynthetic origin of long-chain ascaroside precursors and the precise mechanisms governing functional group assembly on ascaroside derivatives.

In the β-oxidation pathway, functional specialization can be achieved between short-chain molecules (such as acr#3) and long-chain molecules (such as acr#18) [112]. Nematode behavior, such as aggregation, dispersal, Dauer formation, and sex ratio modulation, is controlled by this process. Importantly, β-oxidation integrates energy metabolism with signaling synthesis. For instance, food scarcity promotes the production of ascr#2/ascr#3, which activates the DAF-7/TGF-β pathway to induce Dauer larvae development. Moreover, this pathway mediates dynamic cross-species interactions: bacteria can degrade β-oxidation byproducts to disrupt nematode communication, while plants recognize specific short-chain ascarosides (e.g., ascr#18) to trigger immune responses. Notably, β-oxidation-derived end products (e.g., ascr#1) extend lifespan via the conserved DAF-16/FOXO pathway, highlighting their potential relevance in aging research. From a translational perspective, targeting β-oxidation enzymes (e.g., ACOX-1) to disrupt ascaroside biosynthesis offers novel strategies for nematode control, while the pathway’s metabolic reprogramming mechanisms provide a unique model for studying host–microbe interactions and longevity regulation. Ultimately, β-oxidation serves as a central metabolic hub that integrates chemical diversity, environmental adaptation, and interspecies communication, underpinning nematode ecological competitiveness and survival strategies.

### 6.2. Biosynthesis of Nematode Glucosides

Nematode glucosides are synthesized through the catalytic activity of UDP-glycosyltransferases (UGTs), which transfer a glucose moiety from uridine diphosphate glucose (UDP-glucose) to specific glycosyl acceptors, thereby forming the glucoside backbone structure. Identified acceptor molecules include indole, 3-aminobenzoic acid, phenylacetic acid, pyrrole-2-carboxylic acid, alkanes, isoprenoids, and others [53,108,115]. Although both ascarosides and glucosides are assembled modularly, the sugar donors for glucosides exhibit greater flexibility. This suggests that nematode glucosides may convey more complex and diverse biological signals [116]. UGTs are widely distributed across various nematode species, including parasitic nematodes such as *Haemonchus contortus*, *B. xylophilus*, and *M. incognita* [117]. The *C. elegans* genome encodes 67 UDP-glycosyltransferases, involved in the structural modification of endogenous and xenobiotic metabolites. These glucosylated products are further diversified by carboxylesterases, which regulate nematode behavior, development, and immune responses [118]. Although partial progress has been made in elucidating the biosynthesis of ascarosides and glucosides, their metabolic composition is dynamically influenced by habitat complexity, dietary sources, and nutritional status [119]. This highlights the remarkable adaptability of nematode glucoside production, serving as a critical strategy for nematodes to cope with nutrient fluctuations and environmental toxins.

Glucosylated glycosides are more commonly found in plants, representing a ubiquitous and important class of compound formed by the glycosidic linkage of glucose with other molecules (such as phenolics, steroids, terpenoids, etc.) [120]. These compounds play critical roles in plant growth, development, defense mechanisms, and ecological adaptation, with xenobiotic metabolism serving as a key contributor to the synthesis of glucosidic compounds in both plants and animals [121]. When *C. elegans* or parasitic nematodes such as *Haemonchus contortus* are exposed to a range of xenobiotic toxicants—including benzimidazole drugs (ABZ, mebendazole, fenbendazole, thiabendazole, and oxfendazole)—they rapidly upregulate Phase I and Phase II detoxification enzymes. Notably, a series of glycosyltransferase genes (ugt) are induced, leading to enhanced glucoside synthesis as a detoxification mechanism [122]. In nematodes, the partial inhibition of UGT activity can be achieved using the inhibitor chrysin, while the truncation of glucoside biosynthesis occurs upon the mutation of the transcription factor SKN-1 [123].

Genome-wide association studies (GWASs) have identified carboxylesterases as key enzymes facilitating the formation of ester and amide bonds, playing a pivotal role in mediating modular assembly [53]. Untargeted metabolomics enables the rapid discovery of glucoside derivatives, revealing their biosynthetic origin through the “hijacking” of conserved detoxification mechanisms. Modular metabolites represent an innovative biosynthetic strategy, repurposing cellular waste products to generate novel bioactive compounds that mediate critical biological functions. These metabolites contribute to the structural and functional diversification of pheromones and play essential roles in regulating behavior, development, and aging. Studies on *C. elegans* and other nematodes demonstrate their ability to repurpose biochemical degradation products in complex modular structures with diverse signaling functions. These structures incorporate neurotransmitters (e.g., tyramine, octopamine), indole derivatives, amino acids, nucleotides, and fatty acid metabolites linked to ascarosyl or glucosyl scaffolds. The attachment of diverse acyl groups at the C2 or C6 positions of glucose is mediated by enzymes such as Cel-cest-1.2, Cel-cest-4, and their homologs. The addition of a second acyl group to monoacylglucosides (MOGLs) to form diacylglucosides likely involves additional cest homologs, yielding hundreds of modular ascarosides and glucosides [53]. Wrobel et al. demonstrated that *C. elegans* carboxylesterase Cel-CEST-1.2 and *C. briggsae* carboxylesterase Cbr-CEST-2 collectively produce >150 modular glucosides, with 74 overlapping compounds [124]. This paradigm-shifting biochemical recycling mechanism fundamentally alters our understanding of animal and human metabolism, enabling the continuous generation of novel compounds under variable conditions. By accelerating the structural and functional annotation of unknown metabolites, this discovery opens new research avenues to explore over 100,000 uncharacterized chemical metabolites.

The insulin/IGF-1 and TGF-β signaling pathways promote the biosynthesis of cholesterol into dafachronic acids (DAs). Dafachronic acids—specifically Δ^7^-dafachronic acid (Δ^7^-DA) and Δ^1,7^-dafachronic acid (Δ^1,7^-DA)—serve as ligands for the nuclear hormone receptor DAF-12. Upon activation, DAF-12 governs the decision of whether *C. elegans* enters the Dauer diapause stage and regulates adult lifespan in response to gonadal signaling [125]. Among the dafachronic acids, the biosynthetic pathway of Δ^7^-DA is the most thoroughly characterized. It begins with the desaturation of cholesterol to 7-dehydrocholesterol, catalyzed by the Rieske-like oxygenase DAF-36. Subsequently, a Δ^5^-reductase (yet to be molecularly identified) is proposed in order to convert 7-dehydrocholesterol into lathosterol. This intermediate is then oxidized at the C-3 position by the short-chain dehydrogenase DHS-16 to yield lathosterone. Finally, DAF-9, a cytochrome P450 enzyme, oxidizes the side chain of lathosterone to a carboxyl group, completing Δ^7^-DA biosynthesis. In contrast, the synthesis of Δ^4^-dafachronic acid (Δ^4^-DA) remains poorly understood [126].

## 7. Application of Pheromones in Biological Control

Agricultural pheromones, characterized by their structural conservation, low toxicity, and high specificity, have emerged as a promising eco-friendly pest management approach [4,127]. Currently, insect pheromones are widely applied in agricultural and forestry pest control, while nematode pheromones (e.g., ascarosides) demonstrate considerable potential for developing novel biopesticides.

### 7.1. Pheromone-Based Pest Control

Currently, pheromone-based formulations consist of insect sexual pheromones or semiochemicals. These signals convey various signals relating to areas such as aggregation, foraging, mating, and alarm among insects, serving as their chemical and molecular language for communication. These compounds have been effectively employed as biocontrol agents for targeted pest management against agricultural pests, including the cotton bollworm (*Helicoverpa armigera*), tobacco cutworm (*Spodoptera litura*), and Japanese pine sawyer beetle (*M. alternatus*), with notable efficacy [128]. Furthermore, modern agriculture has integrated chemical ecology principles and CRISPR-based genetic engineering to introduce pheromone synthesis genes into plants, enabling the endogenous production of pest pheromones to recruit natural enemies. For instance, the ectopic synthesis of the aphid alarm pheromone (E)-β-farnesene in *Arabidopsis* enhances the foraging efficiency of aphid parasitoid wasps, while the introduction of (E)-β-caryophyllene synthase genes in corn attracts entomopathogenic nematodes to combat Western corn rootworm (*Diabrotica virgifera*) [129,130]. However, to date, the application of nematode pheromones to plant protection against parasitic nematodes has not yet been widely adopted or commercialized.

Insect sex pheromones have emerged as a cornerstone of modern integrated pest management (IPM) due to their exceptional species specificity, environmental safety, and operational efficiency. As a well-established international monitoring tool, they demonstrate particular efficacy against lepidopteran pests including *Cydia pomonella*, *Grapholita molesta*, and *Spodoptera frugiperda*, providing precise data for timely intervention strategies [131,132]. Their application extends to quarantine systems where tracking population dynamics of invasive species such as *Hyphantria cunea* and *Lymantria dispar* enables proactive risk mitigation. Beyond monitoring, pheromone-based control methods show remarkable field performance: mass trapping techniques effectively suppress populations of *Helicoverpa armigera*, *Tuta absoluta*, and *C. pomonella* through the direct removal of adult stages [133,134,135,136], while mating disruption strategies targeting *G. molesta*, *Chilo suppressalis*, and *Cossus cossus* achieve sustainable control by interfering with reproductive behavior [137,138,139,140]. This dual approach of population monitoring coupled with biologically targeted intervention establishes pheromone technology as both an environmentally benign and scientifically rigorous solution for contemporary agricultural pest challenges.

### 7.2. Pheromone-Based Nematode Control

Biological control ecology views nematode pheromones as an important tool for developing new nematode-based formulations for pest management. Studies have shown that ascaroside mixtures prepared from *Caenorhabditis elegans* can enhance the dispersal of Steinernema feliae infective juveniles (IJs) in Petri dish assays. Further greenhouse experiments revealed that these ascaroside mixtures not only promote nematode dispersal but also improve lethal efficiency against pecan weevils (*Curculio caryae*) and black soldier flies (*Hermetia illucens*) [141]. This suggests that nematode pheromones can enhance the biocontrol potential of entomopathogenic nematodes (EPNs), opening new avenues for their use in insect pest control. Research shows that EPN-infected host extracts, particularly ascaroside pheromones, improve dispersal, and efficacy of infective juveniles (IJs), which are critical factors for biocontrol success [49]. Recently, the U.S. agricultural biotech company Pheronym patented two nematode pheromone-based products—Nemastim™ and Pherocoat they plans to commercialize them for sustainable pest management by manipulating insect behavior [142,143]. Pheronym employs a natural communication platform by utilizing pheromones from commercially available beneficial nematodes to improve their efficacy. These pheromones optimize beneficial nematodes into more efficient predators while simultaneously deceiving parasitic nematodes into perceiving a lack of nearby food sources. This innovation facilitates the broader adoption of organic farming practices by providing an eco-friendly approach to agricultural pest control.

## 8. Challenges and Unresolved Issues in Nematode Pheromone Research

Establishing a comprehensive profile of nematode pheromones is a critical challenge in nematode pheromone research. These pheromones regulate diverse physiological responses, behavioral changes, or phenotypic adaptations at concentrations ranging from femtomolar to nanomolar. Even slight modifications in their chemical structure or functional groups can significantly alter their bioactivity. Traditional isolation methods are often inadequate for purifying individual pheromone compounds, as simple behavioral outputs in organisms are frequently governed by complex pathways or mixtures of signals. In fact, nematodes employ a sophisticated chemical communication strategy where multiple compounds act synergistically in a concentration-dependent manner, exhibiting diverse functional roles. This complexity renders conventional separation techniques unsuitable for pheromone identification. The first ascaroside pheromone, ascr#1, was isolated from 300 L of nematode culture through Dauer formation activity-guided fractionation, yielding only trace amounts [33]. Due to their extremely low secretion levels and functional reliance on synergistic interactions, isolating individual nematode pheromones remains highly challenging. Moreover, commercially available synthetic nematode pheromones—such as ascr#18 and ascr#9, despite their promising biocontrol potential—are prohibitively expensive (costing tens of thousands of RMB per gram), hindering their practical application in agricultural settings.”

Advances in comparative genomics, transcriptomics, metabolomics, and biosynthetic studies, combined with biochemical analyses and CRISPR-Cas9 genome editing, have enabled mechanistic validation in whole organisms [144]. Such integrative approaches facilitate both the discovery of novel metabolite families and the visualization of their biological context through synergistic interactions. By leveraging defective mutants in β-oxidation pathways and correlating their metabolomic profiles with nematode phenotypic variants, researchers have expanded the repertoire of known ascaroside pheromones. NMR-based methods offer notable advantages in linking naturally occurring small molecules to their biological functions, enabling the detection of cooperative signaling and the characterization of chemically labile metabolites. Currently, the dominant approach for pheromone identification is Differential Analysis of NMR Spectra (DANS) [145]. By applying DANS to wild-type (WT) and daf-22 mutant metabolomes, researchers identified ascarosides #6–8, with ascr#8 being the primary male-attracting component. Reverse-phase chromatography coupled with 2D NMR further isolated icas#3 from the daf-22 mutant metabolome, alongside predominant indole ascarosides (icas#3, icas#9, icas#10) and minor derivatives (icas#1, icas#7). Alternatively, capillary electrophoresis–mass spectrometry (CE-MS) has emerged as a complementary technique for profiling polar and charged metabolites in nematodes. For instance, CE-MS analysis of the long-lived daf-2 mutant revealed novel longevity-associated markers—such as nicotinic acid and NAD+—that were undetected by HILIC-MS or NMR-based methods [146].

Modular synthesis offers a distinctive strategy for expanding the nematode signaling molecule library, though key aspects of biosynthetic mechanisms and functional regulation remain unresolved. Comparative metabolomics studies on mutants that are defective in β-oxidation and carboxylase esterase pathways have successfully led to the discovery of new pheromone variants. Beyond the model organism *C. elegans*, exploring non-model nematodes has proven effective for identifying structurally novel metabolites. For instance, an integrated MS/MS and NMR approach enabled the identification of >30 previously unknown compounds in Pristionchus species [147]. Notably, nematodes dynamically remodel their metabolic networks in response to dietary shifts and environmental stressors. Untargeted metabolomics coupled with genomic analyses provides a powerful synergistic approach for annotating these ‘metabolic dark matter’.

In signal transduction research, *C. elegans* serves as a key model for studying pheromones, hormones, nutrient-sensing pathways, and their associated mechanisms. GPCRs (G protein-coupled receptors) mediate the detection of pheromonal cues, activating downstream signaling cascades that drive diverse physiological responses. However, while the nematode genome encodes over 1000 GPCRs, mostly expressed in chemosensory neurons, only a handful of receptor–ligand pairs have been functionally characterized. Similarly, among the 284 nuclear receptors identified in *C. elegans*, few have known activating ligands [148]. Notably, no structural data for any nematode GPCR has been resolved to date, highlighting that the regulatory networks governing pheromone signaling remain largely unexplored.

## 9. Bridging the Lab-to-Field Gap: Practical Challenges and Emerging Frontiers in Ascaroside-Based Nematode Management

While laboratory studies demonstrated the remarkable potential of ascarosides for nematode control [149], translating these findings into practical field applications presents significant challenges. First, the chemical stability of these signaling molecules under variable environmental conditions, such as UV exposure, soil pH fluctuations, and microbial degradation [150], must be optimized to ensure sustained efficacy in diverse agricultural settings. Second, scaling up production while maintaining cost-effectiveness remains a hurdle, particularly for smallholder farmers, as synthetic ascaroside production requires complex organic chemical pathways. Additionally, efficient delivery mechanisms [151], such as controlled-release formulations, soil-embedded microcapsules, or seed coatings [152], must be developed to ensure the precise spatiotemporal distribution of these compounds without the rapid dissipation or off-target effects.

Beyond these translational challenges, key unanswered questions in ascaroside research present exciting avenues for future investigation. For instance, how do soil microbiomes influence ascaroside persistence and function? Could engineered microbes be used as “live delivery systems” to enhance their bioavailability? Furthermore, while some ascarosides trigger plant immunity [19,103], others suppress it. Could antagonists of harmful nematode signals disrupt infestation? Finally, the potential for ascaroside resistance in nematode populations necessitates studies of long-term evolutionary dynamics and adaptive management strategies. Addressing these gaps will not only advance fundamental science but also accelerate the development of a scalable, next-generation nematode control solution.

Studies on nematode-derived ascarosides have unveiled three synergistic strategies for sustainable nematode management: (i) the plant defense activator ascr#18, which primes systemic resistance and metabolic reprogramming in crops, reducing root–knot nematode infections [103,153]; (ii) ascarosides like ascr#7 and ascr#5 that enhance fungal biocontrol by recruiting nematode-trapping fungi, increasing their predatory efficiency [27]; and (iii) seed pretreatment with entomopathogenic nematodes (EPNs) to establish early immune memory in developing plants [154]. Together, these approaches harness the natural signaling systems of nematodes to develop targeted, ecologically sound solutions that simultaneously exploit plant immunity, manipulate nematode behavior, and bolster beneficial soil microbiomes for integrated pest management.

Future research could exploit nematode ascarosides—key pheromone mediators—for targeted biocontrol by manipulating aggregation, dispersal, and mating behaviors. Aggregation signals (e.g., icas#3, icas#9) [38,155] could lure parasitic nematodes into fungal trap zones, while repulsive ascarosides (ascr#5, ascr#10) [40] might shield crops by dispersing pests toward fallow areas. Mating disruption using synthetic analogs (ascr#3, ascr#8) [[34],] could impair reproduction by exaggerating mate-finding or prematurely inducing Dauer states. A coordinated “attract–kill–disrupt” strategy—combining fungal trap nodes, repellent root coatings, and pheromone confusion—could sustainably reduce infestations while minimizing non-target effects. Early field trials must optimize ascaroside stability and delivery (microcapsules, seed coatings) while assessing ecological risks. Addressing these challenges could unlock precision biocontrol tools aligning with integrated pest management, but scalability, cost, and resistance evolution remain critical hurdles. Theoretical frameworks now require empirical validation under real-world conditions to translate lab insights into scalable solutions.

## 10. Conclusions

Chemical communication is a ubiquitous information transfer mechanism in the biological world. It plays a crucial role in regulating the survival, reproductive, defense, and social behaviors of organisms. Ascarosides, a class of multifunctional signaling molecules in nematodes, mediate behaviors such as aggregation and avoidance, reproductive development, and interspecies interactions. They also serve as a key survival strategy for regulating population density and adapting to complex environments through chemosensory perception. Although significant progress has been made in the structural identification of ascarosides and glycosidic pheromones, major challenges remain in the discovery of novel signaling molecules. These challenges include the elucidation of functional mechanisms and the investigation of synergistic effects among mixed signals. Nematode behavior and development regulation often involves the coordinated actions of multiple pheromones. However, the underlying mechanisms of molecular pathways and regulatory networks remain incompletely understood. Future research should integrate multi-omics approaches, including comparative genomics, transcriptomics, and metabolomics, to identify novel signaling molecules in nematodes and systematically elucidate their biosynthetic pathways and regulatory mechanisms. Such efforts will not only deepen our understanding of nematode chemical communication networks but also lay a theoretical foundation for developing environmentally friendly, nematode-targeted biocontrol agents.

## Figures and Tables

**Figure 1 biomolecules-15-00981-f001:**
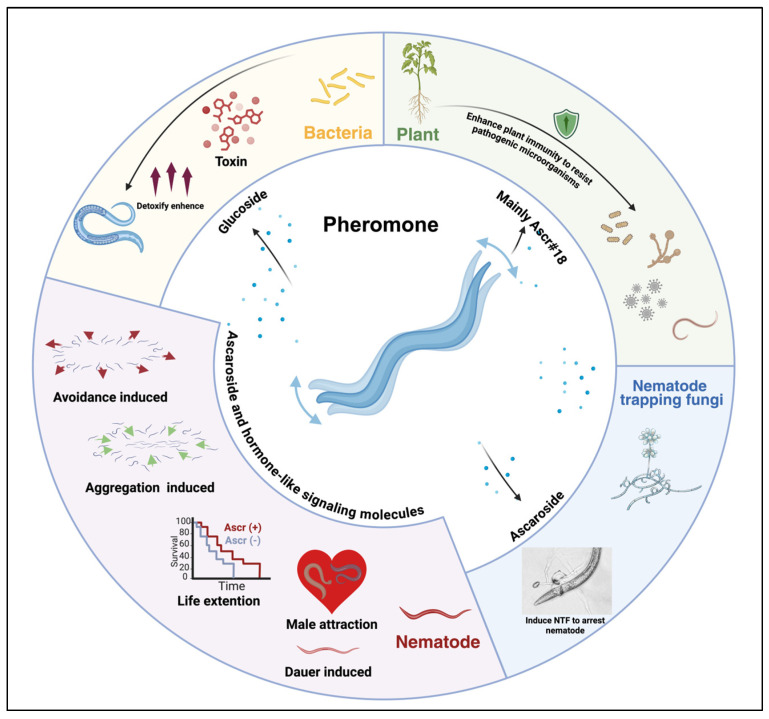
Nematode pheromones as key mediators of behavior, development, and ecological interactions.

**Figure 2 biomolecules-15-00981-f002:**
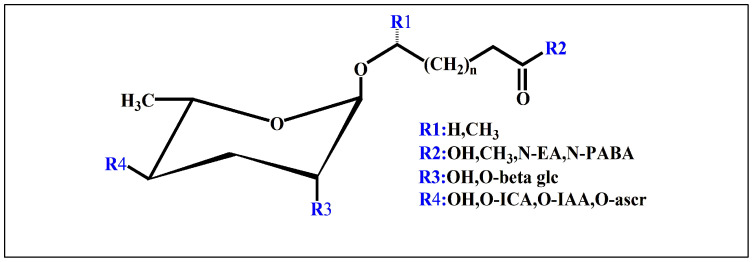
Basic structure of ascarosides.

**Figure 3 biomolecules-15-00981-f003:**
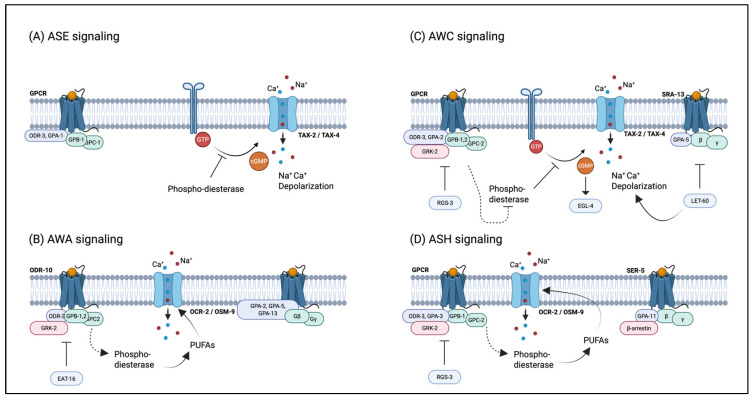
Signal transduction pathways of major chemosensory neurons in *C. elegans.* (**A**) Signaling of the water-soluble attractant neuron ASE; (**B**) signaling of the volatile attractant neuron AWA; (**C**) signaling of the volatile attractant neuron AWC; (**D**) signaling of the volatile attractant neuron ASH.

**Figure 4 biomolecules-15-00981-f004:**
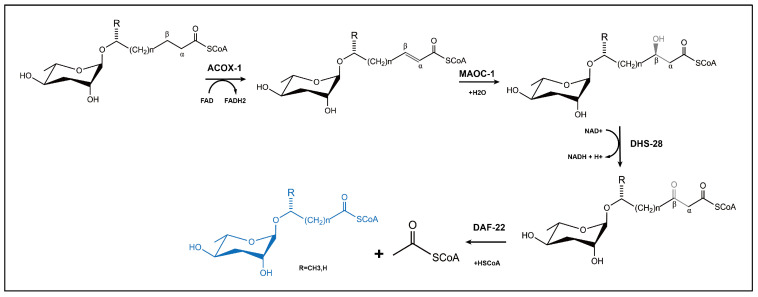
β-oxidation pathway is involved in biosynthesis of ascarosides in nematodes.

**Table 1 biomolecules-15-00981-t001:** Classification and function of nematode pheromones.

**Dauer Pheromone** ascr#1 [33] (very low activity) ascr#2 [34] (high activity) ascr#3 [34] (high activity) ascr#4 [35] (low activity) ascr#5 [36] (high activity) ascr#8 [37] (moderate activity) icas#9 [38] (moderate activity)	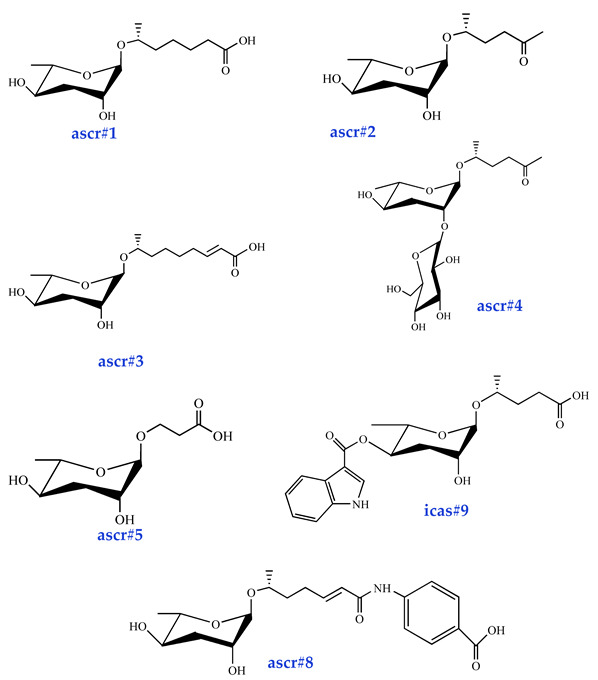
**Sex pheromones** ascr#2 [34] (moderate activity) ascr#3 [34] (high activity) ascr#4 [35] (low activity) ascr#6.1 [37] (moderate activity) ascr#6.2 [37] (activity?) ascr#8 [37] (high activity) ascr#10 [39] (high activity)	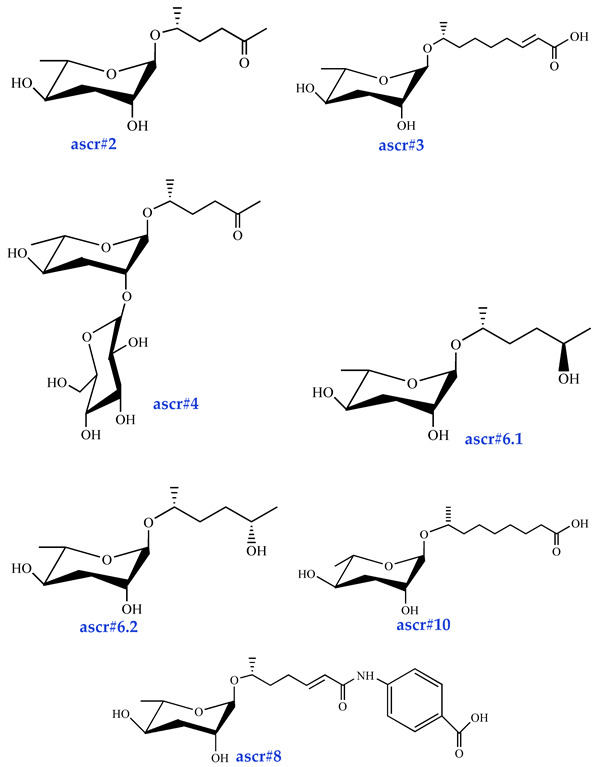
**Aggregation pheromones** icas#3 [16] (high activity) icas#9 [38] (high activity)	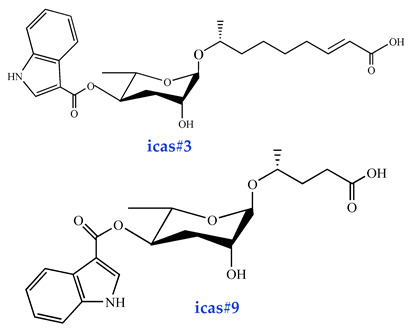
**Behavioral Pheromone** (retention, avoidance, and attraction) ascr#7 [37] ascr#9 [40] ascr#12 [40] ascr#14 [40] ascr#10 [40] ascr#18 [40] ascr#20 [40] ascr#22 [40] ascr#24 [40] ascr#26 [40] oscr#9 [40] oscr#10 [40] oscr#18 [40]	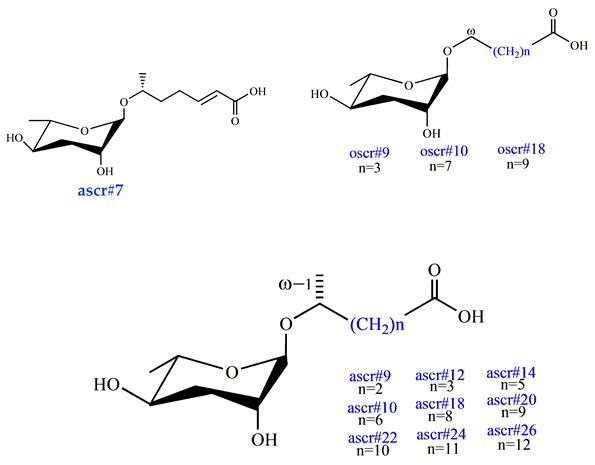
**Glucosides** (detoxification capability) ABZ–glucoside [41] N-(β-D-glucopyranosyl)indole [42] N-(3′-O-phospho-β-D-glucopyranosyl)-indol [42] 1-O-(3′-O-phospho-β-Dglucopyranosyl)-phenazinee [43]	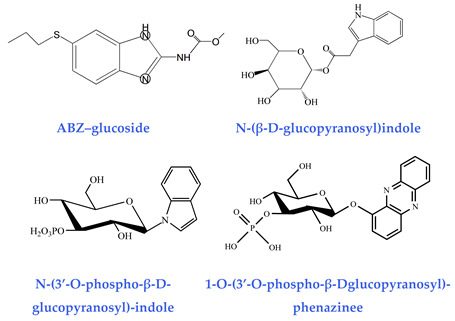
**Non-Glycosidic Pheromones** (regulation of development and behavior) dafachronic acids [44] oleic acids [45] elaidic acids [45] nacq#1 [46]	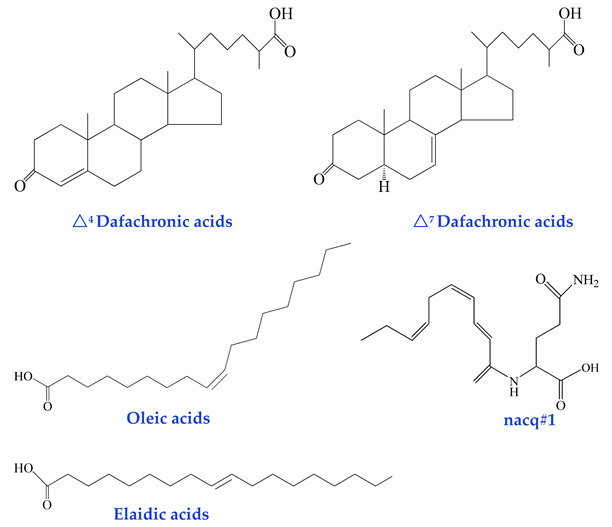

**Table 2 biomolecules-15-00981-t002:** Sensory neurons, receptors, and signal transduction pathways in *C. elegans*.

Neuron	Receptors	G proteins	Signal Transduction	Regulators	Function	Chemical Stimulus
ASE	Receptor guanylate cyclases?	gpa-3	tax-4, tax-2, daf-11, cGMP	osm-9, gpc-1, tax-6, ttx-4, adp-1	Water-soluble chemotaxis	cAMP, cGMP, Lysine, Histidine, Cysteine, Methionine, Biotin, Na^+^, K^+^, Cl^−^
AWA	GPCRs (odr-10)	odr-3, gpa-3, gpa-5, gpa-13; gpa-6	osm-9, ocr-2, fat-3, PUFA, odr-4, odr-8	egl-4, grk-2, tax-6, ttx-4	Volatile chemotaxis, lifespan (minor)	Pyrazine, Diacetyl (low), 2, 4, 5-Trimethylthiazole (low), Butyric acid, Isobutyric acid, Benzyl proprionate, soamyl alcohol (low)
AWC	GPCRs (str-2)	odr-3(major), gpa-3, gpa-2, gpa-5, gpa-13	tax-4, tax-2, daf-11, odr-1, odr-4, odr-8, cGMP	osm-9, eg/-4, grk-2, tax-6, ttx-4, sdf-13, adp-1, let-60, goa-1	Volatile chemotaxis, lifespan, navigation	Diacetyl, 2, 4, 5-Trimethylthiazole (low), Benzyl proprionate, Benzaldehyde, Isoamyl alcohol, Acetone, Dimethylthiazole, 1--Methylpyrrole, 1-Pentanol, 2-Cyclohexylethanol, 2-Ethoxythiazole, Benzyl mercaptan, 2-Heptanone, 3-Pentanedione
AWB	GPCRs	odr-3	tax-4, tax-2, daf-11, odr-1, cGMP	kin-29	Volatile avoidance	Ketones, 2-Nonanone, Serrawettin W2
ASH	GPCRs	odr-3(major), gpa-3(major), gpa-11, gpa-1, gpa-13, gpa-14, gpa-15	osm-9, ocr-2, fat-3, PUFA, qui-1(chemical only), osm-10(osmo only), odr-4	grk-2(chemical, partial osmo)tax-6, ttx-4, gpc-1, kin-29	Nociception: osmotic avoidance, nose touch avoidance, chemical avoidance, social feeding	Acidic pH, Basic pH (>10.5), Copper, Cadmium, SDS, Bitters quinine, Diacetyl (high), Benzaldehyde (high), Isoamyl alcohol (high), 1-Octanol, 2-Nonanone
ASI	GPCRs;	gpa-1, gpa-3, gpa-4, gpa-5, gpa-6, gpa-10, gpa-14	tax-4, tax-2, daf-11, cGMPodr-1, odr-4		Dauer formation, chemotaxis (minor), navigation	cAMP, cGMP, Lysine, Histidine, Na^+^, K^+^, Cl^−^, Biotin, SDS
ADF	GPCRs;	odr-3, gpa-3, gpa-10, gpa-13	osm-9, ocr-2, odr-4		Dauer formation, chemotaxis (minor)	cAMP, cGMP, Na^+^, K^+^, Cl^−^, Biotin, Acidic pH
ASG	GPCRs;	gpa-3	tax-4, tax-2, cGMP, odr-4		Dauer formation (minor), lifespan, chemotaxis (minor)	cAMP, cGMP, Lysine, Histidine, Cysteine, Biotin, Na^+^, K^+^, Cl^−^
ASJ	GPCRs;	gpa-1, gpa-3, gpa-9, gpa-10, gpa-14	tax-4, tax-2, daf-11, daf-21, cGMP, odr-1, odr-4		Dauer formation and recovery, chemotaxis (minor), lifespan	Phenazine-1-carboxamide, Pyochelin, SDS
ASK	GPCRs;	gpa-2, gpa-3, gpa-14, gpa-15	tax-4, tax-2, cGMP, odr-1, daf-11, odr-4	kin-29	avoidance (minor), chemotaxis (minor), lifespan, navigation	Lysine, Histidine, Cysteine, Methionine, Acidic pH, Bitters quinine
ADL	GPCRs;	gpa-1, gpa-3, gpa-11, gpa-15	osm-9, ocr-2, odr-4		avoidance (minor), social feeding	Copper, Cadmium, Isoamyl alcohol (high)
URX, AQR, PQR	gcy-35, gcy-36	gpa-8	tax-4, tax-2, cGMP	npr-1	oxygen/aerotaxis, social feeding	?
PHA, PHB	GPCRs	gpa-1, gpa-2, gpa-3, gpa-6, gpa-9, gpa-13, gpa-14, gpa-15	osm-9, ocr-2, odr-4		Avoidance (antagonistic)	SDS, Dodecanoic acid

## Data Availability

No new data were created or analyzed in this study.

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
