# Peer review of "Nematode Pheromones as Key Mediators of Behavior, Development, and Ecological Interactions"

_biomolecules, 2025, doi:10.3390/biom15070981_

Round 1

Reviewer 1 Report

Comments and Suggestions for Authors

The review has the title “Nematode Pheromones as Key Mediators of Behavior, Development, and Ecological Interactions”, and it encompasses a very systematic and thorough review of nematode pheromones, which includes their classification and functions, regulatory control systems, interspecific signaling, biosynthetic pathways, and possible roles as biological control agents from an ecological perspective.

Suggestions for minor changes to improve the review:

The abstract and introduction outlines challenges in application and gives some potential future research directions, although the body of the text lacks a more thorough discussion that would be appropriate for the concluding section. Elaborating on the practical challenges with converting laboratory work into effective field applications (stability, cost, delivery mechanisms of pheromone-based products) would add value. Equally, addressing the specific unanswered questions or emerging areas within nematode pheromone research could motivate further research.

There are few examples mentioned for the biocontrol potential. This section would benefit from a few more specific (albeit tentative or theoretical) suggestions for several types of pheromones, such as aggregation, dispersal, and mating disruption, in order to enhance their role in pest control management scenarios.

The review looks at interactions from other domains (plant immunity, fungal predation). If the literature allows, elaborating a little more on the molecular details of these interactions would be very interesting and insightful.

Specific comments:

Lines 127-129: “In addition to the intrinsic role of the central dogma 'gene-mRNA-protein', precise editing of chemical communication in terms of both quality and quantity in real-time is crucial for nematodes to respond to complex external changes [51,52].” – This sentence is quite dense with information. It may be clearer to break it up, or at the very least, explain “precise editing of chemical communication".

Author Response

Part A. Reviewer #1’s comments and our responses

Reviewer #1: The review has the title “Nematode Pheromones as Key Mediators of Behavior, Development, and Ecological Interactions”, and it encompasses a very systematic and thorough review of nematode pheromones, which includes their classification and functions, regulatory control systems, interspecific signaling, biosynthetic pathways, and possible roles as biological control agents from an ecological perspective.

Response: Dear reviewer 1, thank you very much for your attention to our manuscript and helping our paper processing. It was a great encouragement to receive your positive and valuable comments. We have made a thoroughgoing revision of the manuscript based on your suggestion. Please refer to the revised manuscript for more details.

Comment 1. The abstract and introduction outlines challenges in application and gives some potential future research directions, althouge body of the text lacks a more thorough discussion that would be appropriate for the concluding section. Elaborating on the practical challenges with converting laboratory work into effective field applications (stability, cost, delivery mechanisms of pheromone-based products) would add value. Equally, addressing the specific unanswered questions or emerging areas within nematode pheromone research could motivate further research.

Response: We strongly appreciate your kind suggestion and add more details about applications in Part 9.

  1. Bridging the Lab-to-Field Gap: Practical Challenges and Emerging Frontiers in Ascaroside-Based Nematode Management

While laboratory studies demonstrated the remarkable potential of ascarosides for nematode control, translating these findings into practical field applications presents significant challenges. First, the chemical stability of these signaling molecules under variable environmental conditions, such as UV exposure, soil pH fluctuations, and microbial degradation, must be optimized to ensure sustained efficacy in diverse agricultural settings. Second, scaling up production while maintaining cost-effectiveness remains a hurdle, particularly for smallholder farmers, as synthetic ascaroside production requires complex organic chemical pathways. Additionally, efficient delivery mechanisms, such as controlled-release formulations, soil-embedded microcapsules, or seed coatings, must be developed to ensure the precise spatio-temporal distribution of these compounds without rapid dissipation or off-target effects.

Beyond these translational challenges, key unanswered questions in ascaroside research present exciting avenues for future investigation. For instance, how do soil microbiomes influence ascaroside persistence and function? Could engineered microbes be used as "live delivery systems" to enhance their bioavailability? Furthermore, while some ascarosides trigger plant immunity, others suppress it. Could antagonists of harmful nematode signals disrupt infestation? Finally, the potential for ascaroside resistance in nematode populations necessitates studies of long-term evolutionary dynamics and adaptive management strategies. Addressing these gaps will not only advance fundamental science but also accelerate the development of a scalable, next-generation nematode control solution.

Comment 2. There are few examples mentioned for the biocontrol potential. This section would benefit from a few more specific (albeit tentative or theoretical) suggestions for several types of pheromones, such as aggregation, dispersal, and mating disruption, in order to enhance their role in pest control management scenarios.

Response: We have addressed this query in section 9:

Future research could exploit nematode ascarosides-key pheromone mediators-for targeted biocontrol by manipulating aggregation, dispersal, and mating behaviors. Aggregation signals (e.g., icas#3,icas#9)could lure parasitic nematodes into fungal trap zones, while repulsive ascarosides (ascr#5, ascr#10)might shield crops by dispersing pests toward fallow areas. Mating disruption using synthetic analogues (ascr#3, ascr#8) could impair reproduction by exaggerating mate-finding or prematurely inducing dauer states. A coordinated “attract-kill-disrupt” strategy—combining fungal trap nodes, repellent root coatings, and pheromone confusion—could sustainably reduce infestations while minimizing non-target effects. Early field trials must optimize ascaroside stability and delivery (microcapsules, seed coatings) while assessing ecological risks. Addressing these challenges could unlock precision biocontrol tools aligning with integrated pest management, but scalability, cost, and resistance evolution remain critical hurdles. Theoretical frameworks now require empirical validation under real-world conditions to translate lab insights into scalable solutions.

Comment 3. The review looks at interactions from other domains (plant immunity, fungal predation). If the literature allows, elaborating a little more on the molecular details of these interactions would be very interesting and insightful.

Response: Exposure of root or leaf tissues to ascr#18, a molecule analogous to microbial PAMPs—rapidly activates pattern-triggered immunity (PTI) via membrane-bound receptors like NEMATODE-INDUCED LRR-RLK1 (NILR1), triggering MAPK cascades, SA/JA signaling, defense-related genes, a defense strategy exploited in Arabidopsis and tomato against microbial pathogens and root knot nematodes.

As your suggestion, we also make progress in section 5:

In two predacious fungi species (A. oligospora and Arthrobotrys flagrans), the mechanism by which G protein-coupled receptors (GPCRs) respond to nematode pheromones to trigger trap formation has been elucidated. In A. flagrans, the GPCR GprC exhibits dual localization on both the plasma membrane and the outer mitochondrial membrane. Mitochondrial GprC enhances respiratory metabolism. Upon nematode presence, GprC binds to the G protein α subunit GasA, initiating downstream signaling to activate fungal predatory behavior….

Once again, thank you very much for your arduous work and instructive suggestions.

Yours sincerely

Reviewer 2 Report

Comments and Suggestions for Authors

The Manuscript ID: biomolecules-3639851 addresses nematode pheromones as key mediators of behavior, development, and ecological interaction. It reviews the mechanism of small molecule signaling substances regulating nematode behavior and development as important tool for developing environmentally friendly biological control agents. It presents nematode pheromones as essential chemicals for signaling intra-species and interspecies communication, regulating development, reproduction, and social behavior. The paper properly highlighted their structural diversity that enables ecological adaptation and cross-kingdom interactions, influencing fungal predation and plant immunity. This review focuses on the classification, function, and regulatory mechanisms of nematode pheromones, interspecific signal transmission, and biosynthesis pathways. The authors stressed the potential of nematode pheromones as environmentally friendly biological control agents as well as the challenges currently encountered in their application.  Thus, the topic is relevant to the field of the study as it cures a specific gap in the field.

The subject is worth publication and the authors did a good job. Yet, further insights might improve the study and specific improvements should be considered:

  • The authors are full aware of the fact that “modern agriculture has integrated chemical ecology principles and CRISPR-based genetic engineering to introduce pheromone synthesis genes into plants, enabling the endogenous production of pest pheromones to recruit natural enemies.” Apart from this genetic engineering and related omics that they could successfully clarify in details, they would better report on current efforts to improve the capacity of plants to activate the innate immunity of plants against plant-parasitic nematodes (PPNs). These latter are the most widespread and damaging soil-borne pathogens/pests. So, it is good to also add to the text applying mixtures of biocontrol agents (BCAs) to mature plants or, more efficiently, to sow seeds in adequately enriched soils to obtain seedlings more resilient to environmental stresses. In such cases they would discuss also interspecies communication implying nematode pheromones for integration of their stated biocontrol approaches against PPNs to boost plant immune system.
  • Genome editing with other tools are stated by the authors to constitute such integrative approaches and facilitate both the discovery of novel metabolite families and the visualization of their biological contexts through synergistic interactions. It is also important to mention some people's reluctance or reservations about the use of genetically engineered crops, especially in human food.
  • Scientific names should be written in tilted form or at least underlined; e.g.: “ elegans lifespan extension…” Should be “C. elegans lifespan extension…”…. “accelerates reproductive development in C. elegans by hastening sexual…” Should be “accelerates reproductive development in C. elegans by hastening sexual…” Also “including the cotton bollworm (Helicoverpa armigera), tobacco cutworm (Spodoptera litura), and Japanese pine sawyer beetle (Monochamus alternatus)…” Should be “including the cotton bollworm (Helicoverpa armigera), tobacco cutworm (Spodoptera litura), and Japanese pine sawyer beetle (Monochamus alternatus)…”

Therefore, I would suggest accepting it after minor revision.

Author Response

Part B. Reviewer #2’s comments and our responses

Reviewer #2: 

The Manuscript ID: biomolecules-3639851 addresses nematode pheromones as key mediators of behavior, development, and ecological interaction. It reviews the mechanism of small molecule signaling substances regulating nematode behavior and development as important tool for developing environmentally friendly biological control agents. It presents nematode pheromones as essential chemicals for signaling intra-species and interspecies communication, regulating development, reproduction, and social behavior. The paper properly highlighted their structural diversity that enables ecological adaptation and cross-kingdom interactions, influencing fungal predation and plant immunity. This review focuses on the classification, function, and regulatory mechanisms of nematode pheromones, interspecific signal transmission, and biosynthesis pathways. The authors stressed the potential of nematode pheromones as environmentally friendly biological control agents as well as the challenges currently encountered in their application. Thus, the topic is relevant to the field of the study as it cures a specific gap in the field.

The subject is worth publication and the authors did a good job. Yet, further insights might improve the study and specific improvements should be considered:

Response: Dear reviewer 2, thank you very much for your attention to our manuscript. The pertinent suggestions you provided are extremely helpful in enhancing the quality of our manuscript. We have made a thoroughgoing revision of the manuscript based on your suggestion. Please refer to the revised manuscript for more details.

Comment 1. The authors are full aware of the fact that “modern agriculture has integrated chemical ecology principles and CRISPR-based genetic engineering to introduce pheromone synthesis genes into plants, enabling the endogenous production of pest pheromones to recruit natural enemies.” Apart from this genetic engineering and related omics that they could successfully clarify in details, they would better report on current efforts to improve the capacity of plants to activate the innate immunity of plants against plant-parasitic nematodes (PPNs). These latter are the most widespread and damaging soil-borne pathogens/pests. So, it is good to also add to the text applying mixtures of biocontrol agents (BCAs) to mature plants or, more efficiently, to sow seeds in adequately enriched soils to obtain seedlings more resilient to environmental stresses. In such cases they would discuss also interspecies communication implying nematode pheromones for integration of their stated biocontrol approaches against PPNs to boost plant immune system.

Response:

That's right, we have added more content about interspecies communication implying nematode pheromones for integration of their stated biocontrol approaches against PPNs to boost the plant immune system,we make progress in section 9:

Studies on nematode-derived ascarosides have unveiled three synergistic strategies for sustainable nematode management: (i) the plant defense activator ascr#18, which primes systemic resistance and metabolic reprogramming in crops, reducing root-knot nematode infections. (ii) ascarosides like ascr#7 and ascr#5 that enhance fungal biocontrol by recruiting nematode-trapping fungi, increasing their predatory efficiency;and (iii) seed pretreatment with entomopathogenic nematodes (EPNs) to establish early immune memory in developing plants. Together, these approaches harness the natural signaling systems of nematodes to develop targeted, ecologically sound solutions that simultaneously exploit plant immunity, manipulate nematode behavior, and bolster beneficial soil microbiomes for integrated pest management.

Comment 2. Genome editing with other tools are stated by the authors to constitute such integrative approaches and facilitate both the discovery of novel metabolite families and the visualization of their biological contexts through synergistic interactions. It is also important to mention some people's reluctance or reservations about the use of genetically engineered crops, especially in human food.

Response: Indeed, we fully agree that there is public reluctance or reservations regarding the safety of gene editing in food or food-related applications.

While this technology shows promise in laboratory settings—for instance, in enhancing nematode pheromone production—its application in field-based biological control requires careful consideration.

Comment 3. Scientific names should be written in tilted form or at least underlined; e.g.: “ elegans lifespan extension…” Should be “C. elegans lifespan extension…”…. “accelerates reproductive development in C. elegans by hastening sexual…” Should be “accelerates reproductive development in C. elegans by hastening sexual…” Also “including the cotton bollworm (Helicoverpa armigera), tobacco cutworm (Spodoptera litura), and Japanese pine sawyer beetle (Monochamus alternatus)…” Should be “including the cotton bollworm (Helicoverpa armigera), tobacco cutworm (Spodoptera litura), and Japanese pine sawyer beetle (Monochamus alternatus)…”

Response: In response to your suggestions, we have revised these inaccurate sections in the manuscript.

Once again, thank you very much for your arduous work and instructive suggestions.

Yours sincerely

Reviewer 3 Report

Comments and Suggestions for Authors

Review report

General comments

This manuscript offers a timely and comprehensive overview of nematode pheromones—tiny chemical messengers with huge implications for ecology, behavior, and sustainable pest management. The authors convincingly advocate for a shift away from conventional nematicides toward biologically inspired alternatives. Notably, the manuscript doesn't just dwell on intra-nematode signaling but casts a much wider net, addressing interspecies, cross-kingdom, and even transgenerational effects of these molecules. The breadth is impressive, and the paper is clearly the product of a well-informed and thoughtful literature survey. With some restructuring and stylistic fine-tuning, this work could become a definitive reference for researchers in nematology, chemical ecology, and bio-based crop protection.

Comments for the Authors

  1. Introduction
  2. The opening drifts too long across general chemical ecology before zeroing in on nematodes. Anchor the relevance of nematodes earlier, ideally in the first paragraph. Why do nematode pheromones matter now, and what’s the practical urgency?
  3. Quantify the impact: What are the crop losses? How effective (or ineffective) are current solutions? What’s the unmet need that this review seeks to address?
  4. Define and distinguish between terms like “pheromones,” “ascarosides,” “chemical cues,” and “semiochemicals.” Avoid confusion for interdisciplinary readers.

  1. Section 2: Chemical Communication in Ecosystems
  2. The current layout jumps between systems (soil vs. air) and kingdoms (plants, microbes, nematodes). Introduce thematic subheadings (e.g., Volatile Signaling, Microbial Interactions, Host-Pathogen Crosstalk) to bring order to the section.
  3. Plant and bacterial signaling examples are informative—but overtake the spotlight. Always tether these examples back to nematodes. What lessons or analogies apply?
  4. Discuss microbial-nematode signaling more directly. How do microbial VOCs or metabolites influence nematode behavior, and vice versa?

  1. Section 3: Definition & Classification of Nematode Pheromones
  2. Lines 130–139 repeat the introduction. Condense and use this section to launch into the actual classification system.
  3. Offer a structured classification—perhaps a figure or table dividing glycosidic vs. non-glycosidic, primary structures vs. derivatives, etc.
  4. Ascarosides dominate the glycoside subsection. Either expand coverage of other glucoside compounds or tighten the treatment of non-glycosidics for symmetry.
  5. Highly detailed chemical info (MS data, genome size stats) could be relocated to supplementary material. Keep the main text conceptually digestible.

  1. Section 4: Functional Mechanisms
  2. Not all readers will have a background in elegans neurogenetics. Consider visual summaries of receptor cascades or behavioral pathways.
  3. Standardize all compound names (e.g., nacq#1, not nacq1). Use IUPAC-style identifiers (ascr#, icas#, etc.) consistently.

Section 5: Interspecies Signaling

  1. Stick with “ascarosides” rather than switching among “ascaroides,” “asc,” or “ascr#” without clarification.
  2. Fungal and bacterial interaction studies are fascinating. Consider ending this section with a brief “key implications” box to summarize how these inter-kingdom signals could be harnessed in agriculture or biotechnology.

Section 6: Biosynthesis

  1. The β-oxidation discussion, while important, needs a visual schematic for accessibility. Prioritize “why it matters” over “how it works” unless the latter is critical for understanding downstream function.
  2. The section doesn’t sufficiently address unresolved biosynthetic questions. A bullet-point list or paragraph on future challenges—like microbial biosynthesis, environmental plasticity, or microbiome-mediated modulation—would add value.

Section 7: Biocontrol Potential

  1. Describe how pheromones enhance EPN behavior. Is it increased IJ activation? Better chemotaxis? Mating interference? A few mechanistic insights would bolster the claims.
  2. The Pheronym examples are great—emphasize these more and suggest future directions like transgenic pheromone production in crops or soil microbiome engineering.

Section 8: Future Challenges

  1. Point out the overreliance on elegans. Emphasize the need to study agriculturally relevant nematodes (Meloidogyne, Bursaphelenchus, Globodera, etc.).
  2. The cost of synthetic pheromones is a key bottleneck. Quantify it (e.g., $/mg), and suggest scalable alternatives (e.g., engineered yeast, synthetic biology).

This is a promising and much-needed review, rich with content and packed with potential. With structural polish and some strategic rebalancing, it can become a high-impact, widely cited paper.

Author Response

Part C. Reviewer #3’s comments and our responses

Reviewer #3: This manuscript offers a timely and comprehensive overview of nematode pheromones—tiny chemical messengers with huge implications for ecology, behavior, and sustainable pest management. The authors convincingly advocate for a shift away from conventional nematicides toward biologically inspired alternatives. Notably, the manuscript doesn't just dwell on intra-nematode signaling but casts a much wider net, addressing interspecies, cross-kingdom, and even transgenerational effects of these molecules. The breadth is impressive, and the paper is clearly the product of a well-informed and thoughtful literature survey. With some restructuring and stylistic fine-tuning, this work could become a definitive reference for researchers in nematology, chemical ecology, and bio-based crop protection.

Response: Dear reviewer 3, we sincerely appreciate your insightful evaluation of our manuscript and your constructive critique regarding its current limitations. We have made a thoroughgoing revision of the manuscript based on your suggestion. Please refer to the revised manuscript for more details.

Comment 1 Introduction

The opening drifts too long across general chemical ecology before zeroing in on nematodes. Anchor the relevance of nematodes earlier, ideally in the first paragraph. Why do nematode pheromones matter now, and what’s the practical urgency?

Quantify the impact: What are the crop losses? How effective (or ineffective) are current solutions? What’s the unmet need that this review seeks to address?

Define and distinguish between terms like “pheromones,” “ascarosides,” “chemical cues,” and “semiochemicals.” Avoid confusion for interdisciplinary readers.

Response: As your suggestion, we have made more progress.

Plant-parasitic nematodes cause over $175 billion in annual crop losses worldwide, yet conventional nematicides often fail due to resistance and environmental harm. This crisis demands precision agrochemicals targeting nematode-specific communication systems—particularly ascarosides, a class of glycolipid signaling molecules that regulate parasitic behaviors through conserved pathways. Structurally diverse ascarosides (e.g., aggregation-promoting ascr#3, immune-triggering ascr#18) function as both intraspecific pheromones and cross-kingdom signals, offering unparalleled leverage for biocontrol by disrupting nematode reproduction, dispersal, and host colonization without broad-spectrum toxicity.

This review synthesizes how ascaroside networks operate from molecular to ecological scales, with emphasis on three unmet needs: (1) translating lab discoveries (e.g., DAF-7/TGF-β-mediated dauer formation) into field tools like pheromone traps or plant-priming formulations; (2) overcoming nematode adaptability through multi-ascarosidal cocktails; and (3) systematizing the search for cryptic signaling molecules in high-impact species (Meloidogyne, Heterodera). By bridging chemical ecology and applied agriscience, we aim to transition nematode management from toxin-dependent paradigms to behaviorally intelligent interventions.

Comment 2. Chemical Communication in Ecosystems. The current layout jumps between systems (soil vs. air) and kingdoms (plants, microbes, nematodes). Introduce thematic subheadings (e.g., Volatile Signaling, Microbial Interactions, Host-Pathogen Crosstalk) to bring order to the section. Plant and bacterial signaling examples are informative—but overtake the spotlight. Always tether these examples back to nematodes. What lessons or analogies apply? Discuss microbial-nematode signaling more directly. How do microbial VOCs or metabolites influence nematode behavior, and vice versa?

Response: The descriptions of bacterial and plant signaling have been removed, and the content now focuses exclusively on nematode pheromone signaling.

Comment 3 Definition & Classification of Nematode Pheromones

Lines 130–139 repeat the introduction. Condense and use this section to launch into the actual classification system.

Offer a structured classification—perhaps a figure or table dividing glycosidic vs. non-glycosidic, primary structures vs. derivatives, etc.

Ascarosides dominate the glycoside subsection. Either expand coverage of other glucoside compounds or tighten the treatment of non-glycosidics for symmetry.

Highly detailed chemical info (MS data, genome size stats) could be relocated to supplementary material. Keep the main text conceptually digestible.

Functional Mechanisms Not all readers will have a background in elegans neurogenetics. Consider visual summaries of receptor cascades or behavioral pathways.

Response: Based on your feedback, we have incorporated the following additions to Table 1: subdividing glycosides into glycosidic vs. non-glycosidic types, as well as categorizing primary structures and derivatives, among other points you indicated.

Comment 4. Functional Mechanisms

Not all readers will have a background in elegans neurogenetics. Consider visual summaries of receptor cascades or behavioral pathways.

Response: To enhance clarity, receptor cascades and behavioral pathways have now been visualized. Furthermore, we have addressed the compound naming consistency throughout the manuscript.

See:

                           Figure 1 Nematode Pheromones as Key Mediators of Behavior, Development, and Ecological Interactions

Comment 5. Interspecies Signaling

  1. Stick with “ascarosides” rather than switching among “ascaroides,” “asc,” or “ascr#” without clarification.
  2. Fungal and bacterial interaction studies are fascinating. Consider ending this section with a brief “key implications” box to summarize could be harnessed in agriculture or biotechnology.

Response: We have added the following boxed text to address your request for a "key implications" summary of agricultural/biotechnological applications:

Nematic Pheromones Mediate Interspecies Interactions

Beyond functioning as self-beneficial compounds regulating nematode development, phenotypic plasticity, and stress resistance, current evidence confirms that nematodes utilize highly conserved ascaroside pheromones for cross-kingdom communication. This primarily occurs in soil ecosystems through nematode-fungi [87], nematode-bacteria [88,89], and nematode-plant interactions [90-92]. Consequently, plants, animals, and microorganisms have evolved conserved genetic mechanisms to perceive and respond to these pheromones through long-term coevolution, playing a pivotal role in maintaining ecological equilibrium in soil communities [93].

Additionally, standardization of "ascarosides" terminology has been implemented throughout the manuscript per your instruction to "Stick with 'ascarosides' rather than switching among 'ascaroides,' 'asc,' or 'ascr#' without clarification."

Comment 6. Biosynthesis

  1. The β-oxidation discussion, while important, needs a visual schematic for accessibility. Prioritize “why it matters” over “how it works” unless the latter is critical for understanding downstream function.
  2. The section doesn’t sufficiently address unresolved biosynthetic questions. A bullet-point list or paragraph on future challenges—like microbial biosynthesis, environmental plasticity, or microbiome-mediated modulation—would add value.

Response: In β-oxidation pathway, functional specialization can be achieved between short-chain molecules (such as acr#3) and long-chain molecules (such as acr#18). Nematode behavior, such as aggregation, dispersal, dauer formation, and sex ratio modulation, is controlled by this process. Importantly, β-oxidation integrates energy metabolism with signaling synthesis. For instance, food scarcity promotes the production of ascr#2/ascr#3, which activates the DAF-7/TGF-β pathway to induce dauer larvae development. Moreover, this pathway mediates dynamic cross-species interactions: bacteria can degrade β-oxidation byproducts to disrupt nematode communication, while plants recognize specific short-chain ascarosides (e.g., ascr#18) to trigger immune responses. Notably, β-oxidation-derived end products (e.g., ascr#1) extend lifespan via the conserved DAF-16/FOXO pathway, highlighting their potential relevance in aging research. From a translational perspective, targeting β-oxidation enzymes (e.g., ACOX-1) to disrupt ascaroside biosynthesis offers novel strategies for nematode control, while the pathway’s metabolic reprogramming mechanisms provide a unique model for studying host-microbe interactions and longevity regulation. Ultimately, β-oxidation serves as a central metabolic hub that integrates chemical diversity, environmental adaptation, and interspecies communication, underpinning nematode ecological competitiveness and survival strategies.

Comment 7: Biocontrol Potential

Describe how pheromones enhance EPN behavior. Is it increased IJ activation? Better chemotaxis? Mating interference? A few mechanistic insights would bolster the claims.

Response: Research shows that EPN-infected host extracts, particularly ascaroside pheromones, improve dispersal and efficacy of infective juveniles (IJs),which are critical factors for biocontrol success.

Oliveira-Hofman, C., Kaplan, F., Stevens, G., Lewis, E., Wu, S., Alborn, H. T., ... & Shapiro-Ilan, D. I. (2019). Pheromone extracts act as boosters for entomopathogenic nematodes efficacy. Journal of Invertebrate Pathology, 164, 38-42.

Once again, thank you very much for your arduous work and instructive suggestions.

Yours sincerely

Round 2

Reviewer 3 Report

Comments and Suggestions for Authors

The authors have fully addressed all of my questions. I have no further comments. I recommend that the manuscript be accepted for publication. Congratulations to the authors on an excellent study and a valuable contribution to scientific progress!